# SEG-LANEDET: 3D LANE DETECTION FROM MONOCULAR IMAGES WITH 2D SEGMENTATION

## ABSTRACT

Monocular 3D lane detection is a fundamental yet challenging task in autonomous driving. Recent advancements primarily rely on constructing 3D surrogates from monocular images and camera parameters. However, misalignment is introduced in current methods due to the lack of dense depth information in datasets, coupled with the inherent depth ambiguity of monocular images. To address this issue, we propose Seg-LaneDet, a simple but effective end-to-end 3D lane detector. We frame the task of 3D lane detection as an elevation from 2D to 3D detection. Specifically, we leverage a pre-trained 2D lane detector to obtain instance segmentation of lanes, of which the segmentation maps serve as the sole prior for the 2D-to-3D module. This allows us to achieve a straightforward 3D lane representation based on front-view segmentation maps. Our method demonstrates performance comparable to state-of-the-art (SOTA) F1 scores on the OpenLane and the Apollo datasets.

## 1 INTRODUCTION

Accurate and robust 3D lane detection is a fundamental and pivotal task in autonomous driving, underpinning critical applications such as lane keeping (Chen & Huang, 2017) and high-definition map construction (Liu et al., 2020). While notable advancements (Luo et al., 2024); (Beijbom et al., 2019) have been achieved through the utilization of various sensors, monocular cameras have attracted considerable attention due to their cost-effectiveness. In contrast to other modalities that often exhibit limitations in texture information, monocular cameras provide abundant information and extended perception, advantageous for detecting slender and flexible lanes.

The lack of depth information is the primary challenge in 3D lane detection from monocular images. The high-quality dense depth data is both prohibitive and scarce, which renders early methodologies (Cohen et al., 2019); (Cai et al., 2022) hard to accomplish. These methods rely on the integration of monocular images with depth estimation.

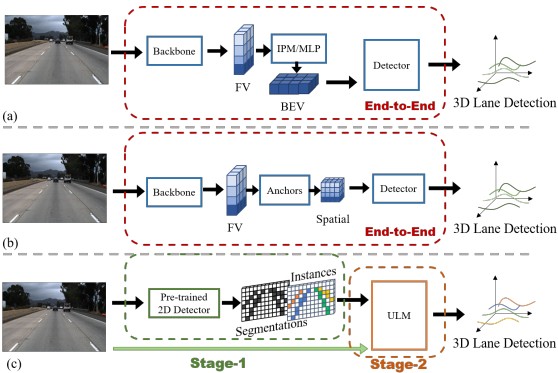

Figure 1: (a) Previous methods mainly utilize inverse perspective mapping (IPM) and Multi-Layer Perceptron (MLP) to transform the features into Bird's Eye View (BEV). (b) Previous methods mainly utilize the front-view (FV) features and designed anchors to regress 3D lanes directly. (c) Our method utilizes a pre-trained 2D detector to get the 2d segmentation and lifts 2D lanes to 3D.

Inspired by advancements in 3D object detection (Belongie et al., 2017); (Carion et al., 2020); (Zhu et al., 2021), contemporary approaches construct 3D surrogates from front-view (FV) images and camera parameters (Dai et al., 2023); (Cao et al., 2023); (Cui et al., 2023); (Bai et al., 2023), thereby obviating the need for depth information and yielding impressive results. Given the strong prior that lanes are consistently situated on the ground, some methods employ Bird's Eye View (BEV) as either the sole or primary 3D proxy, introducing absolute scale and spatial representation. However, trans-

formations of BEV (Bowden et al., 2022), such as Inverse Perspective Mapping (IPM) and Multi-Layer Perceptron (MLP) (Bowden et al., 2022), often lead to misalignments in real-world scenarios, particularly on inclined terrains and uneven surfaces. Alternatively, other methods (Cui et al., 2023); (Bai et al., 2023) aim to directly regress 3D lanes from front-view features and designed anchors, circumventing the non-alignment issues associated with 3D surrogates. Nevertheless, the reliance on dense preset anchors incurs significant resource costs in terms of storage and iteration, making these learning-based strategies less adaptable to data scarcity. We observe that in most driving scenarios, the camera's pose and intrinsic parameters remain relatively fixed relative to the vehicle, resulting in a stable perspective spatial representation in the front view. This stability enables the consistent and reliable mapping of 3D lanes to front-view segmentation maps. Furthermore, 2D segmentation maps provide pixel-level geometric information about lane configurations and strong interpretability, aligning with the robustness and safety imperatives of autonomous driving. Motivated by the 2D-to-3D paradigm prevalent in 3D human pose estimation (HPE), we conceptualize 3D lane detection as an elevation of 2D lane detection, proposing Seg-LaneDet, a **seg**mentation-based model for 3D **lane det**ection. Specifically, we decouple the 3D lane detection process into sub-tasks of lane classification and instance segmentation, integrating these components with the 2D-to-3D lifting task. By leveraging existing 2D lane detection frameworks, our approach constructs a **P**re-trained **T**wo-dimensional Lane **D**etector (PTD) (Honda & Uchida, 2024), which simplifies the classification and instance segmentation while reducing post-processing time during training. To address errors introduced by 3D surrogates, we propose a lane representation predicated on front-view segmentation maps, where the 3D lanes are depicted as pixel-level heatmaps congruent with the segmentation maps. This representation offers the potential for enhanced lane detection resolution. Additionally, we develop a simple yet effective **U**-shaped **L**ifting **M**odule (ULM) (Belongie et al., 2017) and design hierarchical objective functions to further refine our approach.

In summary, our contributions are delineated as follows:

- We introduce Seg-LaneDet, a front-view and anchor-free 3D lane detection framework that leverages 2D lane segmentation maps to achieve precise and efficient 3D lane detection.

- We propose a pixel-level lane representation based on front-view segmentation maps and implicitly express the 3D spatial information through a simple yet effective u-shaped lifting module.

- We conduct extensive experiments on the OpenLane (Chen et al., 2022) and Apollo datasets (Chen et al., 2020) , demonstrating that our proposed method achieves competitive performance in F1 scores.

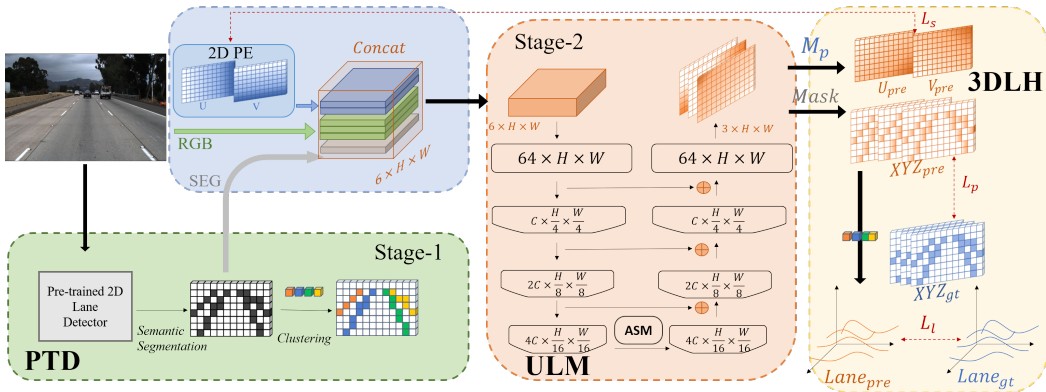

Figure 2: The overall architecture. SegLaneDet is a simple 3D lane detection framework that utilizes front-view segmentation maps. Specifically, the training process consists of two stages. In the first stage, the 2D lane detector is trained on the front-view images to generate the segmentation map. In the second stage, the front-view images, the segmentation maps, and the two-dimensional position embeddings are combined into a $6 \times H \times W$ tensor. This tensor is then processed by the U-shaped lifting module, which transforms it into a $3 \times H \times W$ scene estimate. Finally, a hierarchical loss function constrains the training process across three dimensions: scene, lane, and point.

## 2 RELATED WORK

### 2.1 2D LANE DETECTION

Significant progress has been made in 2D lane detection, widely applied in autonomous driving. These deep learning methods can be categorized into four main approaches: row-wise methods, pixel-wise segmentation, anchor-based methods, and parametric methods. Some methods (Cho et al., 2020); (Chen et al., 2021) set the row anchors in the row direction and the column cells in the column direction to represent the 2D lanes. These methods turn the detection into a classification problem of row vectors. Some works (Li et al., 2022); (Luo et al., 2018); (Cai et al., 2021) consider the 2D lane detection as a segmentation task based on pixel-wise, which has the flexibility. However, the computing cost is more expensive than the row-wise methods. Inspired by region-based object detectors, anchor-based methods (Wang et al., 2018); (Badue et al., 2021b) employ different kinds of anchors to localize lanes. parametric methods (Feng et al., 2022); (Badue et al., 2021a) rethink lane detection as a curve-fitting problem. These methods use priors about lane shapes to represent the parametric representations.

### 2.2 3D LANE DETECTION

To address the growing demand for precise 3D lanes in autonomous driving, researchers have a notable interest in researching 3D lane detection. 3D-LaneNet (Cohen et al., 2019) is the pioneer of the task, which firstly transforms the front-view features to bird-eye-viewed for 3D lane detection. The framework uses two pipelines to fuse information from different views of FV and BEV. Gen-LaneNet (Chen et al., 2020) proposes a two-stage model that first encodes an input image and then only decodes its lane segmentation map. Due to the compression of information compared to the original image, the semantic segmentation graph is difficult to support the 2D-to-3D process. PersFormer (Chen et al., 2022) proposes a novel transformer-based (Vaswani et al., 2017) to realize spatial transformation of features, unifying 2D and 3D lane detection. The other great contribution of PersFormer is the OpenLane dataset, the first large-scale realistic 3D lane dataset. Unlike the above methods, BEV-LaneDet (Cao et al., 2023) utilizes MLP to transform FV features to BEV, which is a simple but effective convolutional neural network (CNN) framework. Furthermore, some studies (Bai et al., 2023); (Dai et al., 2023); (Cui et al., 2023) aim to directly regress 3D lanes from front-view features while incorporating designed anchors with the designed anchors. These methods offer improvements in accuracy, while the dense anchors are needed to mitigate the perspective geometric distortion. Recently, PVALane (Gao et al., 2024) presented a view-agnostic feature alignment architecture, thinning anchors by 2D priors. LaneCPP (Condurache et al., 2024) introduces an elegant representation based on parametric splines.

## 3 METHODOLOGY

As shown in Figure 2, the whole simple network architecture consists of three parts: 1) Pre-trained Two-dimensional Lane Detector (PTD): a pre-trained detector for 2D prior; 2) U-shaped Lifting Module (ULM): Lifting the 2D input to 3D; 3) 3D Lane Head (3DLH): a postprocessing method to obtain the 3D lanes.

Given an input image, our model first detects 2D lanes using a pre-trained 2D lane detector. The output consists of 2D lane instance segmentation, which includes a semantic segmentation map, a clustering map, and a classification. Subsequently, the semantic segmentation map, the corresponding image, and the 2D position embeddings are concatenated and fed into the U-shaped Lifting Module (ULM). Following this, the 3D estimations of the entire scene are processed by the 3D Lane Head. During training, these estimations are handled as 2D position estimates and 3D lane instances; however, during testing, the 2D position estimates utilized for training are not computed.

### 3.1 FRONT-VIEWED REPRESENTATION

The representation of lanes is a critical component of 3D lane detection, necessitating an adaptation to the lane line representation space while ensuring robustness and interpretability. Our approach introduces a 3D lane representation that effectively aligns front-view 2D lanes with their corresponding spatial 3D lanes. Initially, we define 2D lanes as $\mathbf{L}_{2D} = \left\{l_{2D}^i\right\}_{i=1}^{N_l}$, and 3D lanes as $\mathbf{L}_{3D} = \left\{l_{3D}^i\right\}_{i=1}^{N_l}$, with both representations containing $N_l$ lanes. We formulate the $i$-th lane as:

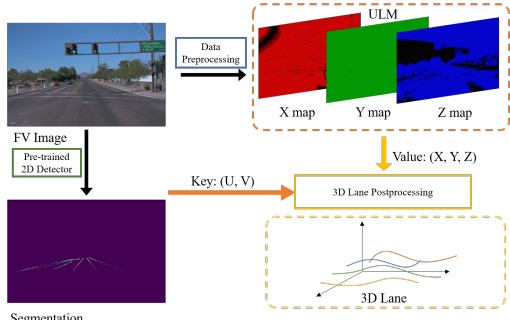

$$l_{2D}^i = \left\{ \left(u^{(i,k)}, v^{(i,k)}, C^i, I^i\right) \right\}_{k=1}^{N_p} \quad (1)$$

where $u^{(i,k)}, v^{(i,k)}$ denote the image coordinates of $k$-th point of the current lane, while $C^i$

Figure 3: An illustration of the front-viewed representation. **Keys** are the lane pixels extracted from the segmentation map. **Values** are stored in XYZ maps generated by ULM.

and $I^i$ represent the class and instance. To leverage the lane classifications and instances from the 2D lane detection output, we represent the 3D lanes as follows:

$$l_{3D}^i = \left\{ \left(x^{(u,v)}, y^{(u,v)}, z^{(u,v)}, C^i, I^i\right) \right\}_{u,v \in l_{2D}^i} \quad (2)$$

where $x^{(u,v)}, y^{(u,v)}, z^{(u,v)}$ denote the ground coordinates of the $k$-th point of the current lane. This simple and ready representation method enables the identification of specific points on a particular lane using the pixel coordinate values derived from the 2D lane detection results. Unlike previous methods, our approach employs dynamic points representing a lane in the front view and the 3D space.

### 3.2 PRE-TRAINED 2D LANE DETECTOR

The solutions for 2D lane detection are relatively well-established. We modify and pre-train a 2D lane detector based on the CLRerNet-Res34 (Honda & Uchida, 2024) to obtain segmentation, classification, and instance results. To incorporate positional information in the image coordinate system, we introduce a 2-dimensional position embedding (Liu et al., 2022), randomly jittering it within the range of $(-0.5, 0.5)$. We concatenate the original image $\mathbf{Img} \in \mathbb{R}^{3 \times H \times W}$, the 2D position embedding $\mathbf{P} \in \mathbb{R}^{2 \times H \times W}$, and the lane segmentation at the channel dimension $\mathbf{S} \in \mathbb{R}^{1 \times H \times W}$. Consequently, the input $\mathbf{I}$ of ULM is obtained as follows:

$$\mathbf{I} = \mathbf{Img} \oplus \mathbf{P} \oplus \mathbf{S} \in \mathbb{R}^{6 \times H \times W} \quad (3)$$

### 3.3 U-SHAPED LIFTING MODULE

Inspired by previous methods (Cui et al., 2023); (Cao et al., 2023), we posit that spatial transformation is more advantageous for reverse-projecting front-view information into ground space than traditional depth estimation techniques. Accordingly, we design a simple lifting module with a U-shaped architecture. Previous studies (Belongie et al., 2017) have suggested that a pyramid structure can leverage different resolutions to balance global and local information, thereby implicitly incorporating scale. Consequently, we utilize a U-Net architecture as the backbone of ULM. Since we do not employ anchors, we construct a lightweight Absolute Scale Module (ASM) based on MLP to introduce absolute scale information. The ULM consists of the U-Net and the ASM, as illustrated in Figure 3.

To incorporate absolute scale information of the scene, we propose a simplified Absolute Scale Module. This module combines the feature map channels from the bottom of the U-shaped structure with the width dimension and performs a fully connected operation on the height. Since lanes in the image extend along the elevation, the absolute scale differences between individual pixels are significant, making them easier to learn.

### 3.4 Losses

To the best of our knowledge, existing 3D lane detection datasets only contain the spatial information of lanes. Therefore, we propose a hierarchical loss function that operates at three levels: scene, lane, and point. Specifically, the point-level loss imposes constraints on the absolute errors of lane positions, the lane-level loss regulates the relative shape of lane lines, and the scene-level loss ensures lifting consistency.

#### 3.4.1 Point-Level Loss

The primary objective of 3D lane detection is to predict every point on the lanes accurately. To achieve this, we apply a mask to the output of ULM, utilizing the segmentation map generated by the 2D lane detector. This approach obtains the 3D coordinates $x, y$, and $z$. Furthermore, the Mean Squared Error (MSE) loss function is employed to quantify the point-level loss:

$$L_{px} = \frac{1}{H \times W} \sum_{i}^{H \times W} 1_{obj} \left( \sigma_x \left( \tilde{x}_i - x_i \right)^2 \right) \tag{4}$$

Where $1_{obj}$ indicates whether the point is masked. $\sigma_x$ denotes the weight of x coordinate, $\tilde{x}_i$ denotes the predict value, and $x_i$ denotes the ground truth. The same formulation applies to the $y, z$ coordinates, resulting in $L_{py}$ and $L_{pz}$.

#### 3.4.2 Lane-Level Loss

Inspired by the motion loss (Lin et al., 2020) utilized in 3D Human Pose Estimation (HPE), we introduce a straightforward yet effective loss function. Specifically, leveraging the front-view representation, we can easily ascertain the lane to which a particular point belongs. To implement the loss, we first compute the cross-product between adjacent points along a single lane for both the predicted and ground truth values. These cross-product results are then compared to characterize the lane shape. The $k$-th predicted point in the $i$-th lane is represented as $p_k^i = [x_k^i, y_k^i, z_k^i]$, and the corresponding ground truth is denoted by $P_k^i = [X_k^i, Y_k^i, Z_k^i]$. The lane-level loss is formulated as follows:

$$L_l = \sum_{i}^{N_l} \sum_{k}^{N_p - 1} \left( p_k^i \times p_{k+1}^i - P_k^i \times P_{k+1}^i \right) \tag{5}$$

#### 3.4.3 Scene-Level Loss

To tackle the challenge posed by the absence of scene-depth information, we draw inspiration from the architecture of autoencoder (AE) (Ng et al., 2011). While the inverse projection from 2D to 3D space is an ill-posed problem, the projection from 3D to 2D space offers a well-defined solution. We project the predicted 3D representation of the entire scene back into image space and compare it with the 2D position embedding. This approach allows us to constrain the consistency of the dimensional ascent, achieving a pixel-level, fine-grained spatial representation. Typically, the projection matrix from 3D to 2D can be derived using the intrinsic and extrinsic parameters, as illustrated in the following equation:

$$z_c \begin{bmatrix} u \\ v \\ 1 \end{bmatrix} = \begin{bmatrix} f_x & 0 & u_0 & 0 \\ 0 & f_y & v_0 & 0 \\ 0 & 0 & 1 & 0 \end{bmatrix} \begin{bmatrix} R & t \\ 0^T & 1 \end{bmatrix} \begin{bmatrix} x_w \\ y_w \\ z_w \\ 1 \end{bmatrix} \tag{6}$$

where $z_c, f_x, f_y, u_0, v_0$ are intrinsic parameters, $R, t$ are extrinsic parameters. $[u, v, 1]^T$ denote the image coordinates, and $[x_w, y_w, z_w, 1]^T$ denote the world coordinates.

However, in practical applications, the camera is frequently in motion, which leads to inaccuracies in the projection results derived from intrinsic and extrinsic parameters, as illustrated in the accompanying figure. To mitigate projection errors, we employ the Direct Linear Transformation (DLT) method to compute the projection matrix. Specifically, we establish a system of linear equations using the 3D points and their corresponding pixel coordinates from the ground truth. We solve for the projection matrix using Singular Value Decomposition (SVD). Given that a sufficient number

of corresponding points can be identified within a single image, we assert that further nonlinear optimization is unnecessary to fulfill the application's requirements.

$$
\begin{bmatrix} u_k^i \\ v_k^i \\ 1 \end{bmatrix} = \begin{bmatrix} p_{11} & p_{12} & p_{13} & p_{14} \\ p_{21} & p_{22} & p_{23} & p_{24} \\ p_{31} & p_{32} & p_{33} & p_{34} \end{bmatrix} \begin{bmatrix} x_k^i \\ y_k^i \\ z_k^i \\ 1 \end{bmatrix} \tag{7}
$$

The equations derived from all pairs of points can be expressed as a homogeneous linear system of equations, ultimately yielding the following form:

$$
A\mathbf{p} = \mathbf{0} \tag{8}
$$

Where $A$ is a $2n \times 12$ matrix, $n$ represents the number of point pairs, and $\mathbf{p}$ is the vector of the flattened projection matrix. We obtain the minimum norm solution of $A\mathbf{p}$ using SVD, and then reconstruct and normalize $\mathbf{p}$ into $3 \times 4$ projection matrix P. The projection matrix obtained through DLT ensures that the projection from 3D to 2D is sufficiently accurate. This accuracy allows us to get precise predicted coordinates $[\tilde{u}, \tilde{v}, 1]^T$ when reprojected back into the image space. Finally, we constructed a scene-level loss function by averaging the L1 distances between the predicted and ground truth:

$$
L_s = \frac{1}{H \times W} \sum_u^H \sum_v^W ((\tilde{u}_{u,v} - u) + (\tilde{v}_{u,v} - v)) \tag{9}
$$

The total loss includes the losses above:

$$
L_{total} = \lambda_p L_p + \lambda_l L_l + \lambda_s L_s \tag{10}
$$

Where $\lambda_p$, $\lambda_l$, and $\lambda_s$ are set to 1.0, 0.1, and 1.0 in our experiments, respectively.

## 4 EXPERIMENT

We evaluate our model on two monocular image 3D lane benchmarks OpenLane (Chen et al., 2022) and Apollo (Chen et al., 2020).

### 4.1 DATASETS AND METRICS

**OpenLane** is the first real-world, large-scale 3D lane dataset, sourcing valuable content from the Waymo dataset (Caine et al., 2020). This dataset comprises 200,000 frames, encompassing various challenging scenarios such as *Up&Down, Curve, Extreme Weather, Intersection, Merge&Split,* and *Night* cases, all at a resolution of 1280×1920. OpenLane includes 880,000 annotated lanes, categorized into a total of 14 distinct types.

**Apollo Synthetic** constructed to stimulate the development and evaluation of 3D lane detection methods, consisting of 10,500 examples from diversified scenarios of highway, urban, and rural environments. The data is split into three subsets, 1) *Standard (simple) scenarios*, 2) *Rare Scenes*, and 3) *Visual Variations*.

**Evaluation Metrics.** We employ the official inspection methods (Chen et al., 2022) to validate our model across the above datasets. The assessment metrics include the F1 score and the X/Z error. Initially, lanes are normalized and subsequently matched using the minimum-cost flow algorithm. A lane is deemed if at least 75% of its points fall within a predefined threshold of 1.5 meters from the ground truth. Furthermore, the error is categorized into close and far errors in the X/Z directions. The close error corresponds to the Y direction within the range of [0, 40] meters, while the far error applies within the range of [40, 100] meters.

### 4.2 EXPERIMENTAL SETTINGS

**Implementation Details.** We utilize an input shape of $360 \times 480 \times 3$ and employ CLRerNet-Res34 (Honda & Uchida, 2024) as our pre-trained 2D lane detector to extract front-view lane segmentation, instance maps, and classifications. Subsequently, we developed a four-layer U-shaped Lifting

Table 1: Comparison with state-of-the-art methods on the OpenLane validation set. "E" denotes end-to-end, and "T" denotes two-stage. **Bold** numbers denote the best results and underlined ones denote the previous best results.

| Methods | Schools | F1 (%)$^{\uparrow}$ | X errors (m)$\downarrow$ | | Z errors (m)$\downarrow$ | |
| --- | --- | --- | --- | --- | --- | --- |
| | | | near | far | near | far |
| 3D-LaneNet (Cohen et al., 2019) | E | 44.1 | 0.479 | 0.572 | 0.367 | 0.443 |
| GenLaneNet (Chen et al., 2020) | T | 32.3 | 0.591 | 0.684 | 0.411 | 0.521 |
| Pers-Former (Chen et al., 2022) | E | 50.5 | 0.485 | 0.553 | 0.364 | 0.431 |
| Anchor3DLane (Dai et al., 2023) | E | 53.7 | 0.276 | 0.311 | 0.107 | 0.138 |
| BEV-LaneDet (Cao et al., 2023) | E | 58.4 | 0.309 | 0.659 | 0.244 | 0.631 |
| LATR (Cui et al., 2023) | E | **61.9** | **0.219** | **0.259** | **0.075** | **0.104** |
| Seg-LaneDet | T | 60.1 | 0.483 | 0.850 | 0.362 | 0.745 |

Table 2: Comparison with state-of-the-art methods on the OpenLane test set under different scenarios. "Mean" denotes the average F1 score of all scenarios. The scenario categories are Up and Down (U&D), Curve (C), Extreme Weather(EW), Night(N), Intersection(I), Merge and Split (M&S). **Bold** numbers denote the best results and underlined ones denote the previous best results.

| Methods | Mean | U&D | C | EW | N | I | M&S |
| --- | --- | --- | --- | --- | --- | --- | --- |
| 3D-LaneNet (Cohen et al., 2019) | 41.7 | 40.8 | 46.5 | 47.5 | 41.5 | 32.1 | 41.7 |
| GenLaneNet (Chen et al., 2020) | 26.4 | 25.4 | 33.5 | 28.1 | 18.7 | 21.4 | 31.0 |
| Pers-Former (Chen et al., 2022) | 47.3 | 42.4 | 55.6 | 48.6 | 46.6 | 40.0 | 50.7 |
| Anchor3DLane (Dai et al., 2023) | 50.1 | 46.7 | 57.2 | 52.5 | 47.8 | 45.4 | 51.2 |
| BEV-LaneDet (Cao et al., 2023) | 53.8 | 48.7 | 63.1 | 53.4 | 53.4 | 50.3 | 53.7 |
| LATR (Cui et al., 2023) | **58.3** | **55.2** | **68.2** | 57.1 | 55.4 | **52.3** | **61.5** |
| Seg-LaneDet | 56.8 | 54.2 | 61.5 | **58.7** | **60.4** | 52.0 | 59.8 |

**Module.** During the encoding phase, each layer of the network halves the feature map size while simultaneously doubling the number of channels. In contrast, the decoding phase reverses this process. Between the encoding and decoding phases, we implement a three-layer MLP to extract features row by row, providing critical cues for absolute scale.

**Training.** All our experiments are trained with batch size 12 and trained models on $4\times$GeForce RTX 4090 GPUs. We optimize the model with the AdamW optimizer with a weight decay of 0.01. We set the learning rate to be $1 \times 10^{-4}$ and use a cosine annealing scheduler. We train the model for 20 epochs on OpenLane and 100 epochs on Apollo.

Table 3: Ablation study on the impact of 2DPE and SEG. "2DPE" denotes the 2D position embedding. "SEG" denotes the segmentation map.

| 2DPE | SEG | F1 (%)$\uparrow$ | X errors (m)$\downarrow$ | | Z errors (m)$\downarrow$ | |
| --- | --- | --- | --- | --- | --- | --- |
| | | | near | far | near | far |
| ✓ | | 6.2 | 0.800 | 1.218 | 0.170 | 1.076 |
| | ✓ | 46.9(+40.7) | 0.597 | 0.936 | 0.283 | 0.703 |
| ✓ | ✓ | 60.1(+13.2) | 0.483 | 0.850 | 0.362 | 0.745 |

Table 4: Ablation study on the impact of ASM. "ASM" denotes the absolute scale module.

| ASM | F1 (%)$\uparrow$ | X errors (m)$\downarrow$ | | Z errors (m)$\downarrow$ | |
| --- | --- | --- | --- | --- | --- |
| | | near | far | near | far |
| | 58.3 | 0.527 | 0.848 | 0.389 | 0.765 |
| ✓ | 60.1(+1.8) | 0.483 | 0.850 | 0.362 | 0.745 |

### 4.3 MAIN RESULTS

We compared our method with five end-to-end state-of-the-art techniques: 3D-LaneNet (Cohen et al., 2019), PersFormer (Chen et al., 2022), Anchor3DLane (Dai et al., 2023), BEV-LaneDet (Cao et al., 2023), and LATR (Cui et al., 2023). We also evaluated our approach against a two-stage method: GenLaneNet (Chen et al., 2020).

**Results on OpenLane.** We present the results on the OpenLane validation set in Table 1, from which it can be seen that Seg-LaneDet achieves 60.1 on the F1 score. Our SegLaneDet outperforms BEV-LaneDet by 1.7 but is 1.8 lower than LATR. SegLaneDet attains the suboptimal result regarding the F1 score, which manifests the superiority of the segmentation-based approach in semanticity. Nevertheless, it must be pointed out that the error performance of our model is unsatisfactory. We

conducted comparisons with other methods and opined that it is caused by the failure to enable the model to learn the absolute scale more clearly.

Additionally, our F1 scores on the test set of OpenLane are excellent, achieving the current best results of 58.7 in *extreme weather*, and 60.4 in *night* scenes. We achieved sub-optimal results of 54.2, 52.0, 59.8, and 56.8 in *up & down*, *intersection*, *merge & split*, and the *mean* test set, respectively. Our results in the *curve* scenario are third best.

Table 5: Table 5: Ablation study on the losses. "$L_p$" denotes the point level, "$L_l$" denotes the lane level, and "$L_s$" denotes the scene level.

| $L_p$ | $L_l$ | $L_s$ | F1 (%)↑ | X errors (m)↓ near | far | Z errors (m)↓ near | far |
|---|---|---|---|---|---|---|---|
| ✓ | | | 51.4 | 0.612 | 0.944 | 0.476 | 0.882 |
| ✓ | ✓ | | 56.2(+4.8) | 0.575 | 0.853 | 0.390 | 0.736 |
| ✓ | | ✓ | 59.8(+8.4) | 0.532 | 0.894 | 0.365 | 0.754 |
| ✓ | ✓ | ✓ | 60.1(+8.7) | 0.483 | 0.850 | 0.362 | 0.745 |

**Results on Apollo.** We present the results on the Apollo dataset in Table 6. We provide a comparison between the previous works and our work. Our work is competitive on the F1 score and the X/Z errors.

Table 6: Comparison with state-of-the-art methods on Apollo 3D Synthetic dataset with three different scenes.

| Scenes | Methods | F1 (%)↑ | X errors (m)↓ near | far | Z errors (m)↓ near | far |
|---|---|---|---|---|---|---|
| Balanced Scene | 3D-LaneNet (Cohen et al., 2019) | 86.4 | 0.068 | 0.477 | 0.015 | 0.202 |
| | GenLaneNet (Chen et al., 2020) | 88.1 | 0.061 | 0.496 | 0.012 | 0.214 |
| | Pers-Former (Chen et al., 2022) | 92.9 | 0.054 | 0.356 | 0.010 | 0.234 |
| | Anchor3DLane (Dai et al., 2023) | 95.4 | 0.045 | 0.300 | 0.016 | 0.223 |
| | BEV-LaneDet (Cao et al., 2023) | 98.7 | 0.016 | 0.242 | 0.02 | 0.216 |
| | LATR (Cui et al., 2023) | 96.8 | 0.022 | 0.253 | 0.007 | 0.202 |
| | Seg-LaneDet | 97.3 | 0.072 | 0.643 | 0.026 | 0.243 |
| Rare Subset | 3D-LaneNet (Cohen et al., 2019) | 74.6 | 0.166 | 0.855 | 0.039 | 0.521 |
| | GenLaneNet (Chen et al., 2020) | 78.0 | 0.139 | 0.903 | 0.030 | 0.539 |
| | Pers-Former (Chen et al., 2022) | 87.5 | 0.107 | 0.782 | 0.024 | 0.602 |
| | Anchor3DLane (Dai et al., 2023) | 97.6 | 0.031 | 0.594 | 0.040 | 0.556 |
| | BEV-LaneDet (Cao et al., 2023) | 99.1 | 0.031 | 0.594 | 0.040 | 0.556 |
| | LATR (Cui et al., 2023) | 96.1 | 0.050 | 0.600 | 0.015 | 0.532 |
| | Seg-LaneDet | 89.9 | 0.173 | 0.517 | 0.042 | 0.762 |
| Visual Variations | 3D-LaneNet (Cohen et al., 2019) | 74.9 | 0.115 | 0.601 | 0.032 | 0.230 |
| | GenLaneNet (Chen et al., 2020) | 85.3 | 0.074 | 0.538 | 0.015 | 0.232 |
| | Pers-Former (Chen et al., 2022) | 89.6 | 0.074 | 0.430 | 0.015 | 0.266 |
| | Anchor3DLane (Dai et al., 2023) | - | - | - | - | - |
| | BEV-LaneDet (Cao et al., 2023) | 96.9 | 0.027 | 0.320 | 0.031 | 0.256 |
| | LATR (Cui et al., 2023) | 95.1 | 0.045 | 0.315 | 0.016 | 0.228 |
| | Seg-LaneDet | 93.5 | 0.084 | 0.574 | 0.047 | 0.274 |

## 4.4 Ablation Study

We conducted extensive experiments on the OpenLane dataset to validate the design of our model. Specifically, we performed ablation studies on data preprocessing, ULM, and losses. All experiments and training maintained consistent configuration settings.

**Data Preprocessing.** We concatenate the original image with corresponding 2D position embeddings and lane segmentation maps of the same size, employing this as a straightforward data augmentation method. The effectiveness has been validated by training it with the individual masking of both the 2D position embeddings and the lane segmentation maps. As shown in Table 3, our SegLaneDet fails to function effectively without the segmentation map input, underscoring that our methodology is fundamentally based on the segmentation maps. Furthermore, the two-dimensional position embedding significantly enhances performance, which improves our model on both sides.

**U-shaped Lifting Module.** Referencing BEVLaneDet, we constructed a scale pyramid encoder and a decoder of the same scale for comparison with our U-shaped Lifting Module (ULM). As

shown in Table 4, the experimental results demonstrate that the ULM outperforms the scale pyramid architecture. ASM played a part in helping SegLaneDet make some gains in F1 scores but sadly did little to improve the absolute X/Z accuracy.

**Losses.** To validate the rationale and effectiveness of our method's loss functions, we experimented with various loss combinations, as detailed in Table 5. The experiment utilizes the point-level loss function, revealing that the lane-level and scene-level loss functions separately result in increases of 4.8 and 8.4 in the F1 score, respectively. When both loss functions are incorporated together, the improvement is 8.7. Our analysis indicates that the internal factors contributing to the enhancements provided by these two loss functions partially overlap, which explains why the combined effect is less than the sum of their individual contributions.

## 5 CONCLUSIONS

In this work, we propose Seg-LaneDet, a front-view and simple 3D lane detector. We innovatively decompose 3D lane detection into two sub-tasks: 2D lane detection and dimensional lifting. Furthermore, we introduce a frontal-view-based representation of 3D lanes, bypassing the need for 3D proxies and cumbersome anchors. Our proposed ULM effectively integrates both relative and absolute scale cues. Additionally, we introduce a hierarchical loss function tailored for the monocular modality. Extensive experiments demonstrate the simplicity and efficacy of Seg-LaneDet. However, our performance on the X/Z error metric is suboptimal, which may reflect a common issue associated with segmentation-based methods. We hope that our work will spark further advancements.

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
