# OpenReview forum: "Seg-LaneDet: 3D Lane Detection from Monocular Images with 2D Segmentation"
_ICLR.cc/2025/Conference — ICLR 2025 Conference Withdrawn Submission_

### Official Review · Reviewer_wvLa · 2024-10-22

**Soundness:** 2
**Presentation:** 2
**Contribution:** 1
**Rating:** 3
**Confidence:** 4

**Summary:**

This paper proposes a two-stage method for 3D lane detection. In the first stage, a pre-trained 2D lane detector identifies lanes from front-view images. In the second stage, a U-shaped network lifts the detected lanes into the 3D scene by estimating their spatial positions. In addition to a point-level loss, the paper introduces lane-level and scene-level loss functions to enhance the accuracy and consistency of lane predictions.

**Strengths:**

This paper leverages 2D lane segmentation maps to enable 3D lane detection by capturing detailed semantic information from the image. It further employs a U-shaped lifting module to estimate depth, transforming 2D segmented lanes into 3D space.

**Weaknesses:**

1. This paper lacks novelty, as the combination of 2D lane segmentation and depth estimation has already been proposed by ONCE-3DLanes [1]. Additionally, ONCE-3DLanes does not rely on pre-trained segmentation; instead, it jointly learns segmentation during the training process. This distinction makes the proposed method simpler and less comprehensive than ONCE-3DLanes.  Furthermore, ONCE-3DLanes is neither adequately discussed in the related work section nor compared in the experimental results. A comparison is strongly recommended to properly position the contribution of the proposed method within the context of prior work.

2. Although the proposed scene-level loss shows promising results in the ablation study (Table 5), it remains unclear how UV supervision benefits the U-shaped Lifting Module (ULM), whose primary objective is depth estimation. The connection between UV coordinate supervision and improved depth prediction needs further explanation.

3. The definition of ASM is unclear. Could you please specify its inputs and outputs to clarify its role and functionality?

[1]@InProceedings{yan2022once,
title={ONCE-3DLanes: Building Monocular 3D Lane Detection},
author={Yan, Fan and Nie, Ming and Cai, Xinyue and Han, Jianhua and Xu, Hang and Yang, Zhen and Ye,
Chaoqiang and Fu, Yanwei and Bi Mi, Michael and Zhang, Li},
booktitle={Proceedings of the IEEE/CVF Conference on Computer Vision and Pattern Recognition},
year={2022}
}

**Questions:**

1. Why adopt two-stage training instead of end-to-end training? If the first stage makes a prediction error, it cannot be corrected in the second stage. Moreover, since the segmentation results are also fed into the U-shaped Lifting Module (ULM), any inaccuracies in the segmentation might negatively impact the depth estimation performance.

2. The description of ASM is unclear. In Line 210, it is mentioned that ASM is shown in Figure 3, but no such reference to ASM can be found in the figure.

3.I guess from Line 213 that the input to ASM is a single row (line) of the 4C × H/16 × W/16 feature map, with the output having the same dimensions as the input. Is my understanding correct? Additionally, the meaning of absolute scale in your paper remains unclear—does it refer to, or is it equivalent to, depth?

---

### Official Review · Reviewer_KBFY · 2024-11-02

**Soundness:** 2
**Presentation:** 3
**Contribution:** 2
**Rating:** 5
**Confidence:** 4

**Summary:**

This work models 3D lane detection problem as a segmentation-and-lifting paradigm, which first utilize a 2D lane detector to produce 2D instance segmentation results, and then utilize a unet to lift the 2D results to 3D space. To achieve this goal, this paper introduce point-level, lane-level and scene-level loss to regulate the laneline learning.

**Strengths:**

I. The paper models 3D lane detection from 2Dsegmentation-and-lifting perspective, which is an intriguing direction for exploration.

II. To learn a better 3D lane representation, this work introduces a hybrid loss to supervise model learning.

**Weaknesses:**

I. Unclear Notation:

1. The notation $M_p$ in Figure 2 lacks an explanation. Provide a brief explanation for it can be helpful.

II. Lack of Comparative Analysis:

The paper does not include a comparison with SALAD[2] (ONCE-3DLanes: Building Monocular 3D Lane Detection), which employs a related approach. Given SALAD's relevance, should consider including such a comparison, which is crucial to highlight the distinct contributions of Seg-LaneDet and to situate its effectiveness within the current body of work.

III. Limited Novelty:

The method’s novelty appears limited due to its reliance on a relatively straightforward UNet-like Module (ULM) for predicting 3D lane outputs. This approach, while practical, does not clearly stand out against recent, more innovative methods in 3D lane detection. This simplicity raises questions about whether it provides substantial advancements or improvements over existing techniques.

IV. Subpar Performance:

The reported performance of the model does not consistently match that of prior state-of-the-art methods. Notably, the **error** is much higher compared to models like LATR, e.g., X-errors: LATR: [0.219, 0.259] versus This Work: [0.483, 0.850], Z-errors: LATR: [0.075, 0.104] versus This Work: [0.362, 0.745]. This raises concerns about the reliability and real-world applicability of Seg-LaneDet.

V. Experiment Analysis:

Table 2 shows that the model achieves better F1 scores in extreme weather and nighttime scenarios. The paper should provide an analysis of the reasons behind this performance, as it would clarify the proposed method’s effectiveness in challenging environments.

**Questions:**

I. The scene loss description (L265-268) is confusing and ambiguous. It states:

  > To mitigate projection error, ..., we establish a system of linear equations using the 3D points and their corresponding pixel coordinates from the ground truth.

 Specifically, are these pixel coordinates derived from a projection using camera parameters? if yes, what is the difference between this from a direct projection of 3D points using camera parameters? if not, then maybe should provide a breif explaination of how these "accurate" pixel coordinates, corresponding to the 3D points, are obtained. This clarification would help distinguish the introduced method from a direct projection approach.

---

### Official Review · Reviewer_6yNZ · 2024-11-03

**Soundness:** 2
**Presentation:** 2
**Contribution:** 3
**Rating:** 6
**Confidence:** 4

**Summary:**

Key Findings of the Paper

1. **Seg-LaneDet Proposal**: The paper introduces Seg-LaneDet, a novel 3D lane detection framework for autonomous driving, based on 2D segmentation and designed for front-view, monocular camera images. This model effectively converts 2D lane detection results to 3D without relying on depth maps or complex 3D proxies.
2. **U-shaped Lifting Module (ULM)**: A central component of Seg-LaneDet, the ULM, lifts 2D data into a 3D space by incorporating both relative and absolute scale information. This approach aims to bypass misalignment challenges and enhance accuracy on varied road surfaces. This is the most valuable design in this paper to the community.
3. **Hierarchical Loss Function**: The paper introduces a multi-level loss function that improves point, lane, and scene-level accuracy, ultimately enhancing the model's performance on F1 scores. However, the model shows limitations in X/Z accuracy, a known constraint in segmentation-based approaches.
4. **Empirical Performance**: Tested on OpenLane and Apollo datasets, Seg-LaneDet achieved competitive F1 scores, notably excelling in challenging conditions like extreme weather and night scenes. It demonstrated improvements in efficiency and error handling over existing state-of-the-art methods but still showed performance limitations on precise X/Z error measurements.
5. **Ablation Studies**: These studies underscore the essential role of 2D segmentation maps and the benefits of adding positional embeddings. The hierarchical loss function and ULM significantly contribute to the system’s performance, validating the design choices.

**Strengths:**

Strengths

1. **Innovative Approach**: The study introduces a unique method for 3D lane detection that relies on 2D segmentation maps instead of dense depth data, reducing complexity and computational cost. By leveraging 2D-to-3D lifting through a U-shaped Lifting Module (ULM), it provides a simpler yet effective approach that improves compatibility with monocular camera inputs, a highly cost-effective option for autonomous driving.
2. **Extensive Empirical Testing**: With testing on large, real-world datasets (OpenLane and Apollo) under varied conditions (night, extreme weather, curves, intersections), the study provides a thorough evaluation. The model’s competitive F1 scores validate its robustness across different scenarios, demonstrating resilience under complex driving conditions.
3. **Modular and Scalable Design**: The model’s reliance on existing pre-trained 2D lane detection modules allows for adaptability and potentially easier integration with new advances in 2D lane detection.
4. **Writing**: The writing is fluent and the figures are illustrative. The related work covers necessary related topics such as different lane methods and their pros and cons. And the figures are illustrative at an idea level, especially the visualization of the method pipeline.

**Weaknesses:**

Weaknesses

1. **Dependency on Pre-trained 2D Detection**: The reliance on a pre-trained 2D lane detector means that Seg-LaneDet’s performance is tied to the quality of this prior module. Any inaccuracies in the 2D segmentation could propagate errors through the 3D lifting process, impacting overall reliability in situations where lane visibility is compromised, such as in heavy rain or sharp curves. Have you analyzed the impact of 2D detection errors on the final 3D output, or tested methods to make the 3D lifting process more robust to inaccuracies in the 2D segmentation?
2. **The advancement to SoTA LATR**:  Although the proposed method is segmentation-based, it would be much better if the author can analyze what is the gap to the SoTA LATR and explain possible reasons on this. It would be great if there is any ablation studies or analyses to understand the key factors contributing to the performance gap between Seg-LaneDet and LATR.
3. **Some typos**: Table 5 has duplicate table names.

**Questions:**

Question

1. Runtime efficiency of the proposed models. Since the model is in a two-stage fashion, I wonder if it will get slower compared to those one-stage models listed in the table 1 and 2. Also, any optimizations or explanation the authors have implemented to mitigate potential speed disadvantages of their two-stage approach would be appreciated.
2. What is the supposed meaning of the output in ULM module? The shape is (H, W, 3) and the author stated that this is XYZ. I am wondering its supervision.

---

### Note · Authors · 2024-11-14

I have read and agree with the venue's withdrawal policy on behalf of myself and my co-authors.